# Vertical transmission of *Leishmania donovani* with placental degeneration in the pregnant mouse model of visceral leishmaniasis

Haruka Mizobuchi[1]*, Junya Yamagishi[2,3], Chizu Sanjoba[1], Yasuyuki Goto[1]

**1** Laboratory of Molecular Immunology, Department of Animal Resource Sciences, Graduate School of Agricultural and Life Sciences, The University of Tokyo, Bunkyo-ku, Tokyo, Japan, **2** International Collaboration Unit, International Institute for Zoonosis Control, Hokkaido University, Sapporo, Japan, **3** Division of Collaboration and Education, International Institute for Zoonosis Control, Hokkaido University, Sapporo, Japan

* mizobuchi@g.ecc.u-tokyo.ac.jp

## Abstract

Visceral leishmaniasis (VL) is a zoonotic disease caused by infection of macrophages by *Leishmania donovani* or *L. infantum*, and exhibits symptoms such as fever, anemia, and hepatosplenomegaly. VL during pregnancy has been reported to have negative effects such as miscarriage and vertical infection, but the mechanism is not clear. Here, we aimed to establish a pregnant VL mouse model and elucidate its immunopathology. Female BALB/c mice mated 6 months after *L. donovani* infection showed reduced pregnancy rates. The fetus was removed by caesarean section on the 18th day of pregnancy, and *Leishmania* parasite DNA was detected from fetal spleens and livers. As a result, the PCR positive rate was 68.9% (71/103 fetus), and vertical transmission was suspected in 66.7% of infected mothers (12/18 dams). Immunohistochemistry in the fetal livers detected cells positive for the *Leishmania* antigen, kinetoplastid membrane protein 11 (KMP11). In addition, pathological analysis of the VL placenta revealed trophoblast cell atrophy and vasodilation accompanied by CD3⁺ cell infiltration in the infected group. On the other hand, few KMP11⁺ cells were observed in the placenta of the infected group. Furthermore, RNA-Seq analysis revealed that IFN signal activation and cellular immune suppression were induced in the placenta of the infected group. These results suggest that VL in pregnancy induces suppression of placental cellular immunity through IFN and collapse of the placental barrier through trophoblast degeneration, leading to vertical transmission. Because few infected macrophages were observed in the placenta, it is possible that free *Leishmania* parasites in the blood contribute to transmission across the placenta.

**Data availability statement:** The authors confirm that all data underlying the findings are fully available without restriction. All relevant data are within the paper and its Supporting Information files.

**Funding:** This work was supported by Japan Society for the Promotion of Science (23K14109 and 24KJ0080 to HM; 21H02722, 22H05057 and 24K02271 to YG) and Japan Agency for Medical Research and Development (JP223fa627001 to HM). The funders had no role in study design, data collection and analysis, decision to publish, or preparation of the manuscript.

**Competing interests:** The authors have declared that no competing interests exist.

## Author summary

Visceral leishmaniasis (VL) during pregnancy can cause miscarriage and vertical transmission and is an important problem to solve. However, the immunopathology of pregnant VL remains unclear. Our study not only demonstrated reduced pregnancy rates and vertical transmission in pregnant VL mice, but also elucidated the immunopathology of the placenta, the maternal-fetal interface. It is suggested that IFN contribute to cellular immunosuppression and collapse of the placental barrier in the VL placenta, leading to vertical transmission. Importantly, placental degeneration and vertical transmission occurred even though few infected cells were observed in the placenta. The results indicate that the maternal immune response to *Leishmania* infection, rather than the direct interaction with *Leishmania* parasites, induces immunopathology and degeneration of the placenta and contributes to vertical transmission. Pregnant VL can have diverse effects on mother and child depending on complex factors such as the parasite species, host resistance, timing of infection, and gestation period. Based on our pioneering work, it is expected that further research on the pregnant VL immunopathology will be developed in the future.

## Introduction

Visceral leishmaniasis (VL) is a zoonotic disease caused by infection with *Leishmania donovani* (Ld) and *L. infantum*, and is transmitted by the female sandfly, a blood-sucking insect. In VL, *Leishmania* parasites multiply in the spleen and liver, causing symptoms such as fever, anemia, and hepatosplenomegaly, which can be fatal if untreated. It causes an estimated thousands of deaths and 50,000–90,000 new cases each year, mainly in Brazil, East Africa, and India [1]. *Leishmania* parasites invade the host during blood feeding by sandflies and infect macrophages (MΦs) in the spleen and liver. When the MΦ finally ruptures due to the proliferation of the parasites, the parasites released outside the cell infect other nearby MΦs and further proliferate. During the process, a small number of parasites are also released into the bloodstream [2], and the sandfly feeds on the infected host's blood and ingests the parasites, completing the transmission cycle [3].

Because *Leishmania* parasites reside within the host MΦs, cellular immunity generally plays an important role in VL control. Natural killer cells (NK cells) and type 1 T helper cells (Th1 cells) secrete IFN-γ to induce activation of inflammatory MΦs, which kill intracellular parasites by increasing nitric oxide [4–6]. On the other hand, humoral immunity does not play a central role in eliminating *Leishmania* parasites, and B cells and Th2 cells induce anti-inflammatory MΦ activation and contribute to parasite proliferation and worsening of pathology [7,8]. Therefore, research and development of VL immunotherapy that strongly induces cellular immunity is progressing [9]. However, both cellular immunity and humoral immunity do not have a simple exclusive or binary relationship, but interact each other in a complex manner during the pathogenesis of VL [10].

During pregnancy, changes in the maternal immune system increase susceptibility to parasitic infections. Additionally, infection during pregnancy can cause inflammation, anemia, and malnutrition in the mother, which can have a negative impact on the pregnancy [11]. The exact number of pregnant VL cases is unknown due to a lack of epidemiological data. On the other hand, a simple calculation of the estimated annual number of VL cases (50,000–90,000) multiplied by the proportion of pregnant women in endemic areas suggests that approximately 2,000–4,000 cases of VL in pregnancy occur annually [12–14]. Because a study in Naples reported that the incidence of VL in pregnancy (1/83,000) was slightly higher compared to the incidence of VL in adults (1/100,000) [15], the actual number of pregnant VL cases may be higher than our estimate. However, as actual case reports are very limited compared to the estimated number of VL cases in pregnancy, most of the patients are likely to be overlooked. The reason for the gap between the estimated and the reported number is assumed to be a lack of patient awareness due to the difficulty in distinguishing between pregnancy and VL symptoms, and rarely visitation to medical institution in the case of early miscarriage.

VL in pregnancy can affect maternal and fetal health and cause miscarriage [16,17]. It has also been reported that *Leishmania* parasites are vertically transmitted from VL mothers to their fetuses [17–20]. In North America, vertical transmission, rather than sandfly-borne transmission, has been shown to be an important cause of the spread of canine VL [21]. Immune tolerance to the fetus during pregnancy suppresses maternal cellular immunity while inducing an immune state dominated by Th2 cells and regulatory T cells. Conversely, excessive activation of cellular immunity during pregnancy can lead to miscarriage [22,23]. Even though the anti-inflammatory immune status in VL theoretically favors pregnancy maintenance, VL increases the risk of miscarriage. This is thought to be because pregnancy-induced cellular immunosuppression favors *Leishmania* survival [24,25].

Antimony agents and miltefosine, which are used to treat VL, have strong side effects and are not recommended for pregnant patients because they have been reported to have fetotoxicity and a risk of miscarriage [26,27]. Furthermore, immunotherapy that induces excessive cellular immunity has a risk of causing miscarriage or premature birth. Therefore, in order to control VL without negative impact on pregnancy, it is important to understand the immunopathological mechanism. However, most publications regarding VL in pregnancy are case reports, and its immunopathology remains largely unknown. Additionally, although some reports suggest that *Leishmania* parasites are vertically transmitted to the fetus *via* the placenta [28–30], the placental immunopathology in VL has not been elucidated. Therefore, in this study, we established a VL-in-pregnancy mouse model using Ld-infected BALB/c mice and elucidated the effects of chronic maternal VL on maternal-fetal immunopathology.

## Materials and methods

### Ethics statement

The animal experiments were reviewed and approved by an institutional animal research committee and an institutional committee on genetically modified organisms at the Graduate School of Agricultural and Life Sciences, The University of Tokyo (Approval No. P22-117). The experiments were performed in accordance with the Regulations for Animal Care and Use of the University of Tokyo, which were based on the Law for the Humane Treatment and Management of Animals, Standards Relating to the Care and Management of Laboratory Animals and Relief of Pain (the Ministry of the Environment), Fundamental Guidelines for Proper Conduct of Animal Experiment and Related Activities in Academic Research Institutions (the Ministry of Education, Culture, Sports, Science and Technology) and the Guidelines for Proper Conduct of Animal Experiments (the Science Council of Japan).

### Experimental infection

Female BALB/c mice at the age of 6–7 weeks were purchased from Japan Clea, Tokyo, Japan, and were acclimated for 1 week. All mice (3–5 mice per cage) were maintained under specific pathogen-free conditions in a temperature- and

humidity-controlled room under a 12h light/dark cycle with unrestricted access to food and water. *L. donovani* promastigotes (MHOM/NP/03/D10; a gift from the National BioResource Project at Nagasaki University [31]) were cultured in medium 199 (Thermo Fisher Scientific, Waltham, MA, USA) supplemented with 10% heat-inactivated fetal bovine serum (Thermo Fisher Scientific), 100 U/ml penicillin and 100 µg/ml streptomycin (Wako, Osaka, Japan) at 25°C. Experimental infection was performed as previously described [32]. Briefly, Ld promastigotes in late log or stationary phase were washed with PBS by centrifugation at $1,600 \times g$ for 10 min and were resuspended with PBS at the concentration of $1 \times 10^8$ cells/ml. Mice were infected with $1 \times 10^7$ Ld promastigotes by intravenous injection into the tail vein.

## Reproduction and hematological analysis

Twenty-four weeks after infection, infected or naïve female mice were mated with healthy male mice. Day 1 of pregnancy was determined when the mucous plug was detected, and plug-positive female mice were transferred to individual boxes and weighed once every two days.

On the 18th day of pregnancy, blood was collected using the heparinized capillary tubes (TERUMO, Tokyo, Japan) and hematocrit (Ht) was determined by centrifuging the tubes at 15,000 x$g$ for 10 min. For quantitative analyses of blood cells, the number of blood cells were counted by microscopic examination. The peripheral leukocytes were counted with Türk's solution (Merck Millipore, Darmstadt, Germany).

## Dissection of pregnant Ld-infected mice

The gestation was interrupted on the 18th day of pregnancy. Whole blood was collected by cardiac puncture under anesthesia with 4% isoflurane vapor (Pfizer Japan Inc., Tokyo, Japan), and spleen and liver were collected from each dam. After euthanizing the dams, the fetuses and placentas were collected. The total number of fetuses, implantation sites, and embryonic resorptions were counted, and the weight of the fetuses were measured. After decapitating the fetuses under 4% isoflurane anesthesia, their spleens and livers were collected.

## Blood biochemical tests and measurement of serum cytokines

Serum was collected after centrifugation for 10 min at $2,000 \times g$. Blood urea nitrogen (BUN), creatinine (CRE), aspartate aminotransferase (AST), and alanine aminotransferase (ALT) in serum were measured by Oriental Yeast Co., Ltd. (Tokyo, Japan) using the clinical chemical analyzer, BioMajesty JCA-BM6050 (JEOL Ltd., Tokyo, Japan). The concentration of serum IL-4, IL-6, IL-10, IL-17A, TNF-α and IFN-γ were measured by using commercial sandwich ELISA kit (Thermo Fisher Scientific). Absorbance was measured using SpectraMax Paradigm microplate reader (Molecular Devices, LLC., San Jose, CA, USA), and data was analyzed using SoftMax Pro 6 (Molecular Devices, LLC.).

## Evaluation of the parasite burden

Stamp smears of the spleens and livers were fixed for 5 min in methanol and stained for 25 min with 5% Giemsa solution (Merck). Amastigotes were counted by microscopic observation of the stained smear at $1,000 \times$ magnification, and Leishman-Donovan Units (LDU) were enumerated as the number of amastigotes per 1,000 host nuclei times the tissue weight in grams as performed in a previous study [33].

## Extraction of DNA and quantitative PCR analysis

DNA was extracted from fetal spleens and livers using the DNeasy Blood & Tissue Kit (QIAGEN, Hilden, Germany). Quantitative PCR assay was carried out using 2 µl of DNA as the template and 10 µl of SYBR Select Master Mix (Thermo Fisher Scientific) on QuantStudio 5 real-time PCR system (Thermo Fisher Scientific), using the primers specific for large subunit ribosomal ribonucleic acid (LSU-rRNA) gene of *Leishmania* parasites (forward: 5′-GGC GGG CAA CGA AGT GCA

AGA AT-3′ and reverse: 5′-GCA CAC TCC AAC GCA ACC CAC GG-3′). The thermal cycling conditions for the PCR were 95°C for 10 min for polymerase activation, followed by 40 cycles of 95°C for 15 s for denaturation, and 60°C for 1 min for extension. Estimates of fetal Ld numbers were calculated using quantitative PCR by fitting the Ct values of DNA samples extracted from whole fetal livers to a standard curve constructed from known numbers of cultured Ld parasites.

## Histochemical analysis

Placentas and fetal livers were fixed with 20% formalin neutral buffer solution (Wako) and embedded in paraffin. HE staining and immunohistochemical staining was performed as previously described [34]. For immunohistochemical staining, paraffin-embedded tissues, sectioned at 6 μm thickness, were dewaxed and boiled in Tris-EDTA buffer (10 mM Tris Base, 1 mM EDTA-2Na, 0.05% Tween 20, pH 9.0) or sodium citrate buffer (10 mM sodium citrate, 0.05% Tween 20, pH 6.0) for 20 minutes. Endogenous peroxidase activity was blocked by incubating the sections in 3% hydrogen peroxide in methanol for 30 min at room temperature. After blocking, rabbit anti-kinetoplastid membrane protein-11 (KMP11) antibody, rabbit anti-CD206 antibody (Cell Signal Technology, Danvers, MA, USA), goat anti-MRP14 antibody (Santa Cruz Biotechnology, Santa Cruz, CA), goat anti-CD3 antibody (Santa Cruz Biotechnology), rat anti-B220 antibody (BD Biosciences, San Jose, CA), goat anti-ICAM1 antibody (Santa Cruz Biotechnology) or goat anti-VCAM1 antibody (Santa Cruz Biotechnology) was applied to the serial sections of tissues. After washing with PBS, sections were incubated with HRP-conjugated anti-rabbit, rat, or goat IgG (Nichirei Bioscience, Tokyo, Japan). Enzymatic color development was performed using DAB (Nichirei).

For double staining of Dolichos Biflorus (DBA)-lectin and PAS, deparaffinized placental tissues were incubated with 1% α-amylase (Wako) for 20 min. After inactivation of endogenous peroxidase and blocking, the tissues were incubated with biotinylated anti-DBA lectin (Sigma, St Louis, MO, USA) for 1h, and followed by incubation with ExtrAvidin-Peroxidase (Sigma) for 30 min. Enzymatic color development was performed using DAB (Nichirei). Following DAB-lectin staining, PAS staining was performed on the same sections using a kit (Muto Pure Chemicals Co. Ltd., Tokyo, Japan). Briefly, the tissue was incubates with 1% periodic acid for 10 min, and followed by incubation with Schiff's reagent for 30 min at 37°C. Then, the color was developed by incubating with 0.5% sodium bisulfite solution for 10 min. The tissues were counterstained with Mayer's hematoxylin solution (Wako) for 30 s and rinsed in running tap water for 20 min.

The labyrinth zone tissue area excluding the vascular lumen was measured using an all-in-one fluorescence microscope BZ-X800 and its analyzer (KEYENCE, Osaka, Japan) at 200× magnification. For quantitative analyses of infiltrating cells in the placental labyrinth zone, the number of cells was counted in 5 random microscopic fields at 400× magnification.

## Transcriptome analysis by RNA-Seq

Placenta was homogenized with 1 ml TRIzol (Thermo Fisher Scientific) and φ1.0 glass beads in a 2 ml tube using Micro Smash MS100R (TOMY, Tokyo, Japan) at 4°C. After transferred to a 1.5 ml tube, 200 µl chloroform was added and centrifuged at 12,000×g for 15 min at 4°C. The supernatant was mixed with 0.5 ml 2-propanol and centrifuged at 12,000×g for 15 min at 4°C. After washing with ethanol, total RNA was dissolved in UltraPure distilled water (Thermo Fisher Scientific). The quality and quantity of the purified RNA were assessed using a Bioanalyzer (Agilent, Santa Clara, CA, USA). RNA Sequencing libraries were prepared with the TruSeq stranded mRNA Library kit (Illumina, San Diego, CA, USA) and paired-end RNA sequencing using Illumina NovaSeq 6000 System (paired-end reads of 101 bp and a total of 45,589,660 reads) was performed by Macrogen (Tokyo, Japan). The criteria for differentially expressed genes were defined as follows: adjusted $p$ value < 0.05 and the absolute value of the $\log_2$ converted signal > 0.58 to extract genes whose fold change has increased or decreased by 1.5 times or more.

## Statistical analysis

Statistical analysis was performed using GraphPad Prism 10.2 software package (GraphPad Software Inc., San Diego, CA). Results are presented as mean ± standard error of the mean (SE). The differences between the groups of mice

were analyzed by two-way ANOVA followed by Sidak multiple comparisons test. Student's *t* test was used to compare the differences in the results from two independent groups. The difference between the Kaplan-Meier curves between the two groups was tested using the Log-rank test and Chi-square test was used to analyze the conception rate. *P* value less than 0.05 were considered significantly different.

## Results

### Pregnancy does not worsen the parasite burden, anemia, and splenomegaly in Ld-infected dams

First, to evaluate the effect of pregnancy on maternal VL symptoms, hematological analysis and spleen and liver weight measurements of Ld-infected dams were performed on day 18 of pregnancy. As we previously reported [32], Ld infection induced a decrease in peripheral blood Ht levels and RBC counts in nonpregnant controls (Fig 1A). Pregnancy also reduced Ht levels and RBC numbers in the naïve group, but the reduction was more gradual than that of non-pregnant

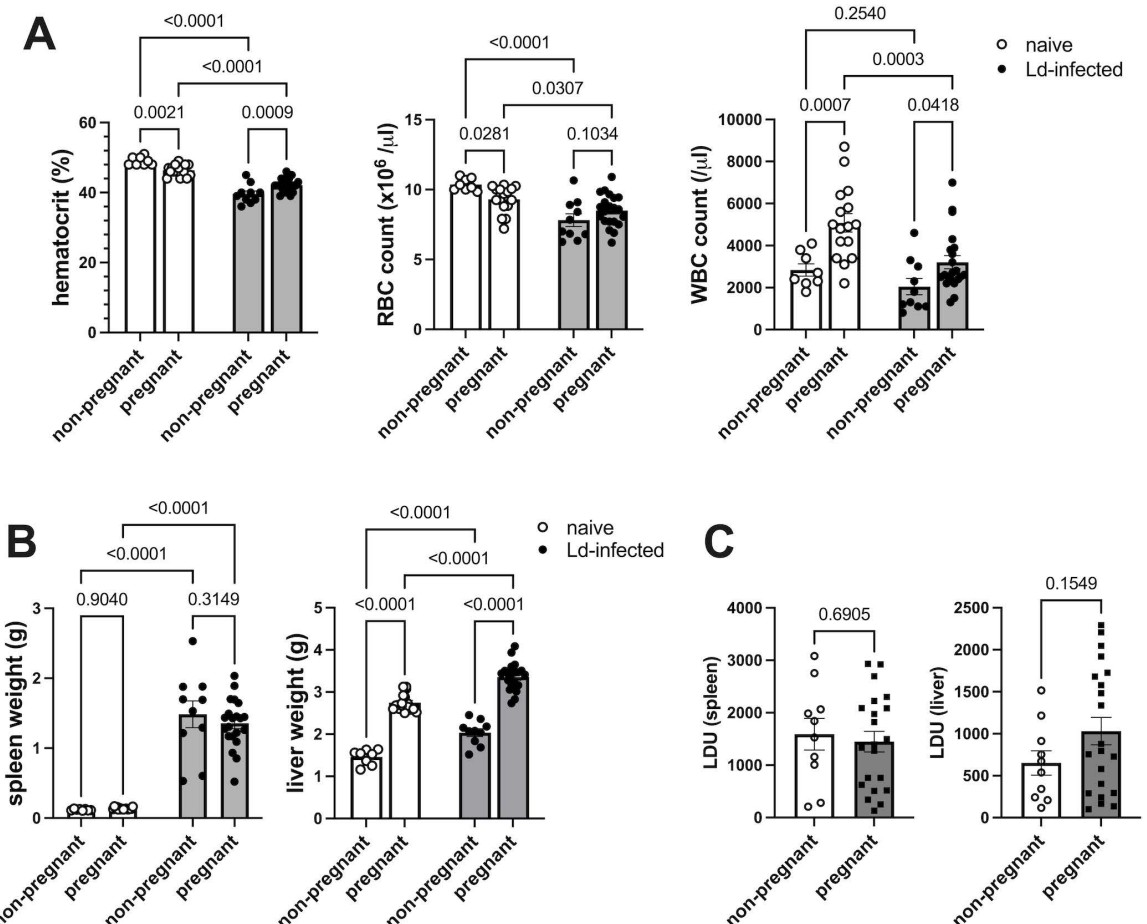

**Fig 1. Pregnancy does not affect maternal parasite burden and Ld-induced anemia and splenomegaly. (A)** Peripheral blood hematocrit, RBC count, and WBC count of naïve and Ld-infected dams on the 18th day of pregnancy (n = 8-21). Pregnancy does not worsen Ld-induced anemia. **(B)** Spleen and liver weights of naïve and Ld-infected dams on the 18th day of pregnancy (n = 8-21). Pregnancy does not worsen Ld-induced splenomegaly. **(C)** Leishman-Donovan Units (LDU) in the spleen and liver of naïve and Ld-infected dams on the 18th day of pregnancy (n = 10-21). Pregnancy does not affect parasite burden in the spleen and liver of Ld-infected dams. Open circles indicate the naïve group, and closed circles indicate the Ld-infected group. Means ± SE are presented. *P* values for Student's t test or two-way ANOVA with Sidak multiple comparisons are shown.

Ld-infected mice. On the other hand, the Ht levels and RBC counts in the pregnant Ld-infected group did not decrease compared to the non-pregnant infected controls, but rather showed a slight tendency to increase due to pregnancy, although it showed lower values than that of the pregnant naïve group. The WBC count increased with pregnancy in both the naïve and Ld-infected groups, but the WBC count was lower in the pregnant Ld-infected group compared to the pregnant naive controls. The results indicate that pregnancy does not worsen anemia caused by Ld infection.

In addition, there was no significant difference in the spleen weight of Ld-infected mice between the non-pregnant and pregnant groups, indicating that pregnancy does not affect splenomegaly caused by Ld infection (Fig 1B). On the other hand, an increase in liver weight was observed in pregnant mice in both the naïve and infected groups, as a normal response to pregnancy. Furthermore, pregnancy did not change LDU (an indicator for evaluating the number of Ld parasites) in the spleen and liver (Fig 1C), although liver LDU in the pregnant group tended to increase slightly as liver weight increased. The results indicate that pregnancy does not worsen the parasite burden, anemia, and splenomegaly in Ld-infected dams.

### Decrease in maternal serum BUN and IFN-γ due to pregnancy

In addition to anemia and splenomegaly, renal and hepatic disorders have also been reported as symptoms of VL. Therefore, we evaluated the effects of pregnancy on serum renal function markers (BUN, CRE) and liver function markers (AST, ALT) in Ld-infected dams. In non-pregnant mice, serum BUN and ALT tended to increase due to Ld infection (Fig 2A). On the other hand, pregnancy suppressed the Ld infection-induced BUN increase. Other marker levels did not change due to pregnancy, and BUN and ALT in the pregnant Ld-infected group were comparable to those in the pregnant naïve group. The results showed that pregnancy does not worsen serum marker levels of kidney and liver damage, which are symptoms of VL, but rather decreases Ld infection-induced serum BUN.

Next, the effect of pregnancy on circulating inflammatory cytokines in infected dams was analyzed. The result showed that serum levels of IFN-γ, IL-4, and TNF-α were increased in the non-pregnant infected group (Fig 2B). On the other hand, pregnancy significantly suppressed Ld infection-induced serum IFN-γ, although it did not affect IL-4 and TNF-α levels. Besides, serum IL-6, IL-10, and IL-17A were also measured, and they were all below the detection limit in all groups. The results indicate that pregnancy reduces Ld infection-induced serum IFN-γ.

### Decrease in mating success rate and pregnancy rate due to Ld infection

Next, the effect of Ld infection on pregnancy was evaluated. The negative rate of vaginal plugs over time after mating was significantly delayed in the Ld-infected group (Fig 3A), meaning that it took longer to detect a vaginal plug in the Ld-infected group. Furthermore, compared to the naïve group, fewer mice in the Ld-infected group became pregnant even when a vaginal plug was confirmed, indicating a significantly lower pregnancy rate (Table 1). On the other hand, there were no differences in body weight changes and weight gain rates of dams with successful pregnancies between the naïve and Ld-infected groups (Fig 3B). In addition, Ld infection did not affect the number of implantations, resorption rate, number of fetuses, fetal weight, or placental weight (Fig 3C). Furthermore, fetal number was not correlated with maternal spleen-liver weight and maternal spleen-liver LDU (Fig 3D). The results indicate that Ld infection reduces mating success and pregnancy rates, although Ld infection has little effect on fetal number or fetal development after pregnancy at least in this model.

### Ld vertical transmission to the fetus.

To assess the effect of maternal Ld infection on vertical transmission to the fetus, whole DNA was extracted from the fetal spleen and liver, and *Leishmania* DNA was analyzed by quantitative PCR. The LSU rRNA gene of the *Leishmania* parasites was selected as the target gene. As a result, LSU rRNA gene was detected in the fetal spleen and/or liver in 66.7%

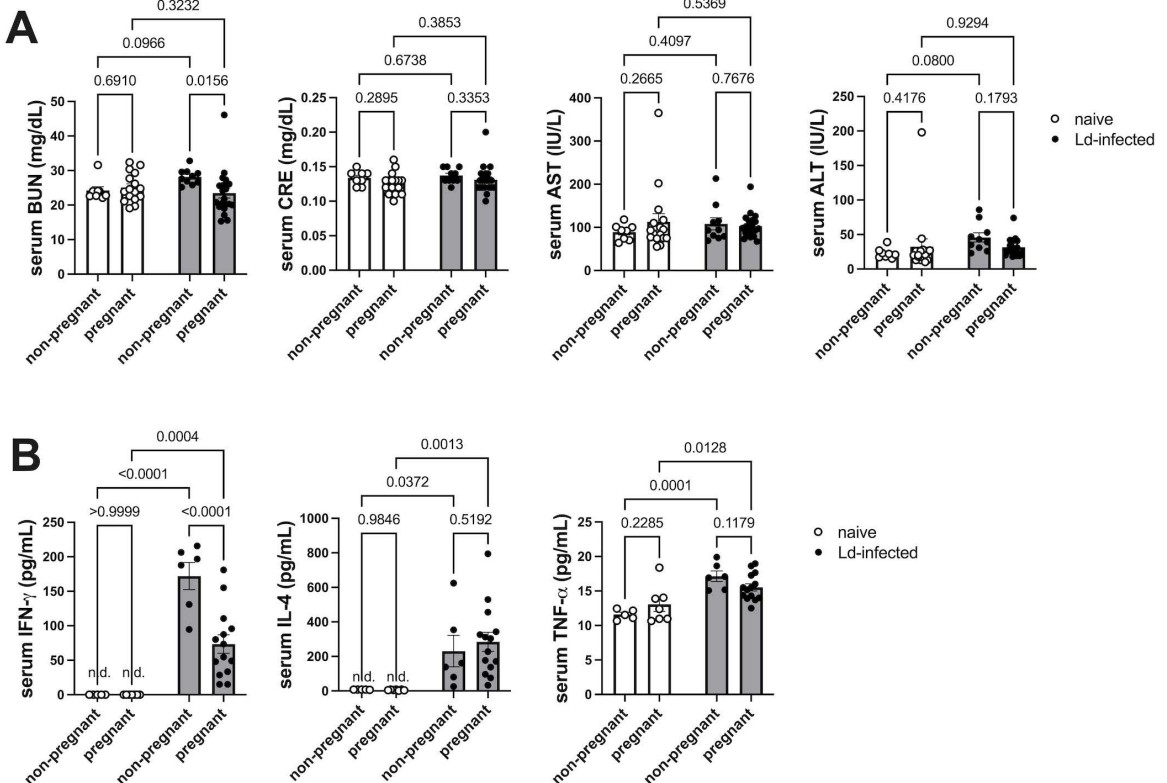

**Fig 2. Pregnancy-induced decrease in maternal serum BUN and IFN-γ. (A)** Serum BUN, CRE, AST, and ALT levels in dams on day 18 of pregnancy (n = 8-21). Pregnancy reduced Ld-induced serum BUN elevation. **(B)** Serum IFN-γ, IL-4, and TNF-α levels in dams on day 18 of pregnancy (n = 5-14). Pregnancy reduced Ld-induced serum IFN-γ elevation. Open circles indicate the naïve group, and closed circles indicate the Ld-infected group. Means ± SE are presented. *P* values for two-way ANOVA with Sidak multiple comparisons are shown. n.d., not detected.

of Ld-infected dams (12/18 dams) (Table 2). Fetal PCR positivity rates were 56.3% in the spleen (58/103 fetuses) and 63.1% in the liver (65/103 fetuses). In some cases, the *Leishmania* gene was detected in both the spleen and liver, and in other cases it was detected only in one of these organs. The total PCR positive rate of fetuses in which the *Leishmania* gene was detected in either the spleen or liver was 68.9% (71/103 fetuses). Additionally, fetal PCR positive/negative results tended to be the same within littermates. The results suggested Ld vertical transmission to the fetus. As a result of quantitative PCR, the average number of Ld parasites in vertically transmitted fetuses was estimated to be 84,720 in the spleen and 160,173 in the liver (Fig 4A).

Based on these results, we performed immunostaining for the *Leishmania* antigen KMP11, and KMP11+ cells were detected in the fetal liver (Fig 4B). KMP11+ cells were detected in the fetal livers of Ld-infected mice (4/11 mice), with the cell numbers ranging from 1 to 2 per section. This is the first report to histologically demonstrate Ld vertical transmission in a VL mouse model. On the other hand, KMP11+ cells were infrequently detected in the placentas of Ld-infected mice, observed in 3 out of 21 samples with 1–3 cells per section. The result demonstrated vertical transmission in our VL mouse model.

## Placental degeneration with T cell infiltration due to Ld infection

Based on the above results, the HE-stained placenta was pathologically analyzed. Histologically, the mouse placenta consists of the maternal decidua, the labyrinth zone (LZ), which is a maze of villi, and the junctional zone where the decidua

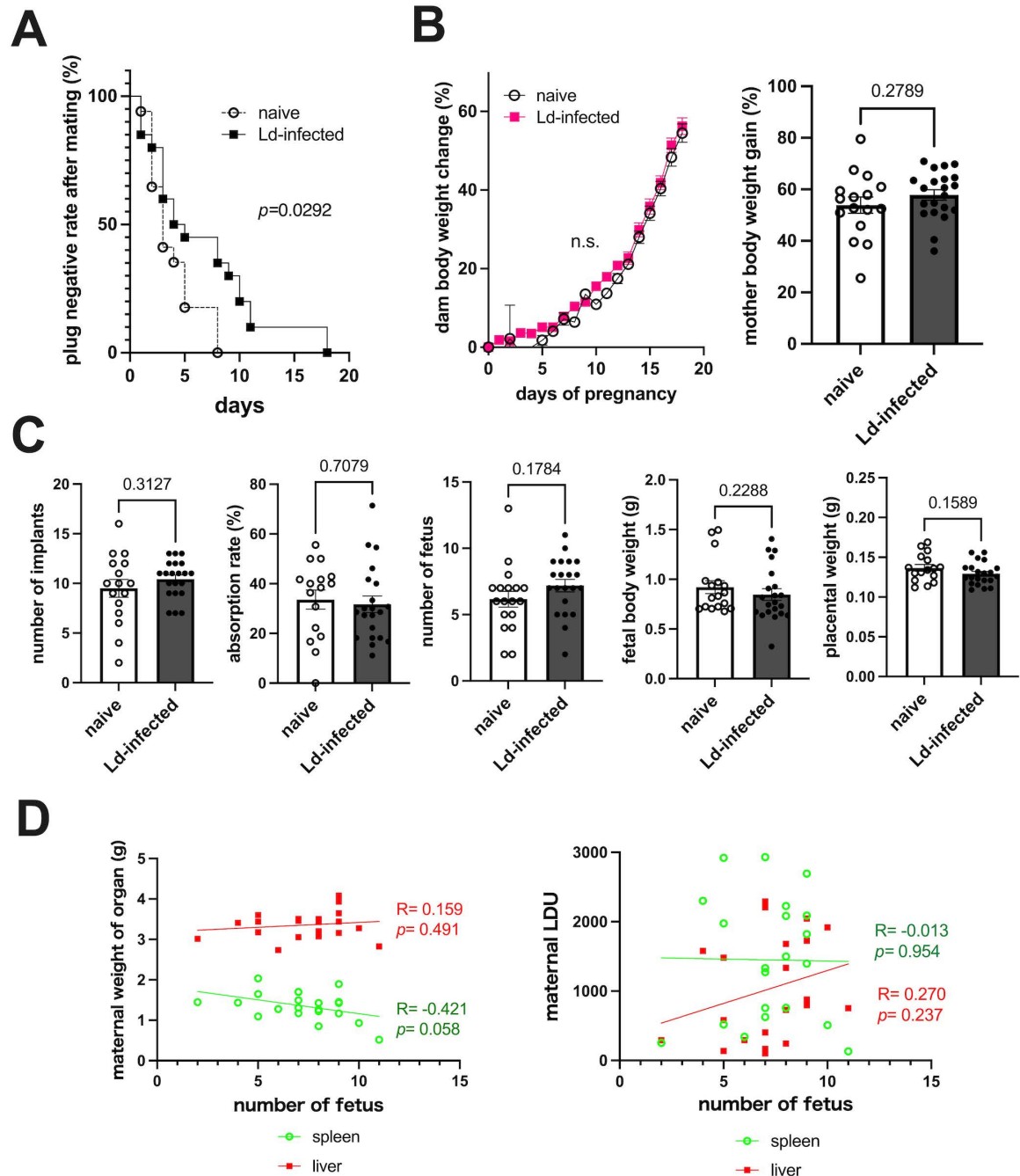

**Fig 3. Reduced mating success rate due to Ld infection. (A)** Changes in the negative rate of vaginal plugs after the start of mating (n = 17-21). Ld-infected mice require a longer mating period than naive controls to detect vaginal plug. **(B)** Left: Change in maternal body weight after pregnancy (n = 16-21). Right: Weight gain rate of the dams on day 18 of pregnancy. Maternal body weight changes after pregnancy are not affected by Ld infection. Individuals who miscarried during the process are excluded. **(C)** Number of implants, resorption rate, number of fetuses, fetal weight, and placental weight on the 18th day of pregnancy (n = 16-21). For fetal weight and placental weight, the average of each dam's offspring is plotted. None of the parameters were affected by Ld infection. **(D)** Left: Relationship between fetal number and maternal spleen/liver weight (n = 21). Right: Relationship between fetal number and maternal spleen/ liver LDU (n = 21). No correlation was observed between fetal number and maternal VL severity. Means±SE are presented. *P* values for Log-rank test, Student's t test or two-way ANOVA with Sidak multiple comparisons are shown. n.s., not significant (*p* > 0.05).

**Table 1. The pregnancy rate.**

|  | Naïve | Ld-infected | Total |
|---|---|---|---|
| **pregnant** | 13 | 14 | 27 |
| **failed pregnancy** | 2 | 12 | 14 |
| **total** | 15 | 26 | 41 |

Odds ratio = 5.571, *p* = 0.0442

**Table 2. PCR positive rates of offspring.**

| dam ID | PCR positive spleen | PCR positive liver | PCR positive offspring |
|---|---|---|---|
| 1 | 0/3 | 0/3 | 0/3 |
| 2 | 0/3 | 0/3 | 0/3 |
| 3 | 5/6 | 6/6 | 6/6 |
| 4 | 0/8 | 1/8 | 1/8 |
| 5 | 0/4 | 0/4 | 0/4 |
| 6 | 0/4 | 4/4 | 4/4 |
| 7 | 4/4 | 4/4 | 4/4 |
| 8 | 0/3 | 0/3 | 0/3 |
| 9 | 5/5 | 5/5 | 5/5 |
| 10 | 9/9 | 9/9 | 9/9 |
| 11 | 1/7 | 7/7 | 7/7 |
| 12 | 7/7 | 7/7 | 7/7 |
| 13 | 5/6 | 6/6 | 6/6 |
| 14 | 11/11 | 11/11 | 11/11 |
| 15 | 5/5 | 5/5 | 5/5 |
| 16 | 6/9 | 0/9 | 6/9 |
| 17 | 0/2 | 0/2 | 0/2 |
| 18 | 0/7 | 0/7 | 0/7 |
| total | 58/103 | 65/103 | 71/103 |

and LZ transition [35]. Trophoblast cells constitute the placental barrier that separates maternal-fetal blood in the LZ. In the placentas of Ld-infected dams, the density of the LZ was lower than that of naive controls, and trophoblast cell atrophy and vasodilation were observed (Fig 4C).

Then, cellular infiltration of the placental LZ was evaluated by immunostaining. MΦs and NK cells are known to be the main populations of blood cells in the normal placenta [36]. The placental MΦ population includes CD206+ fetal MΦs, also called Hofbauer cells, and MRP14+ maternal MΦs [37]. Additionally, the placental NK cell population includes uterine NK cells (uNK) and peripheral blood-derived conventional NK cells (cNK), both of maternal origin. The two NK populations can be distinguished by double staining of DBA lectin staining and PAS staining: uNK is DBA+ PAS+, cNK is DBA- PAS+ [38]. Therefore, quantitative analysis of placental infiltrating cells was performed by immunostaining for CD206, MRP14, B220 (B cell marker), and CD3 (T cell marker) and double staining with DBA/PAS. The results revealed that CD206+ MΦ and MRP14+ MΦ decreased, while CD3+ T cells increased in the placental LZ of the Ld-infected group (Fig 5A). The number of B220+ B cells was not affected by Ld infection. CD3+ T cells in the placentas of the Ld-infected group showed focal infiltrates, which corresponded to areas with high expression of adhesion molecules ICAM1 and VCAM1 in serial sections (Fig 5B), suggesting that the adhesion molecules are involved in CD3+ T cell infiltration. Regarding NK cells,

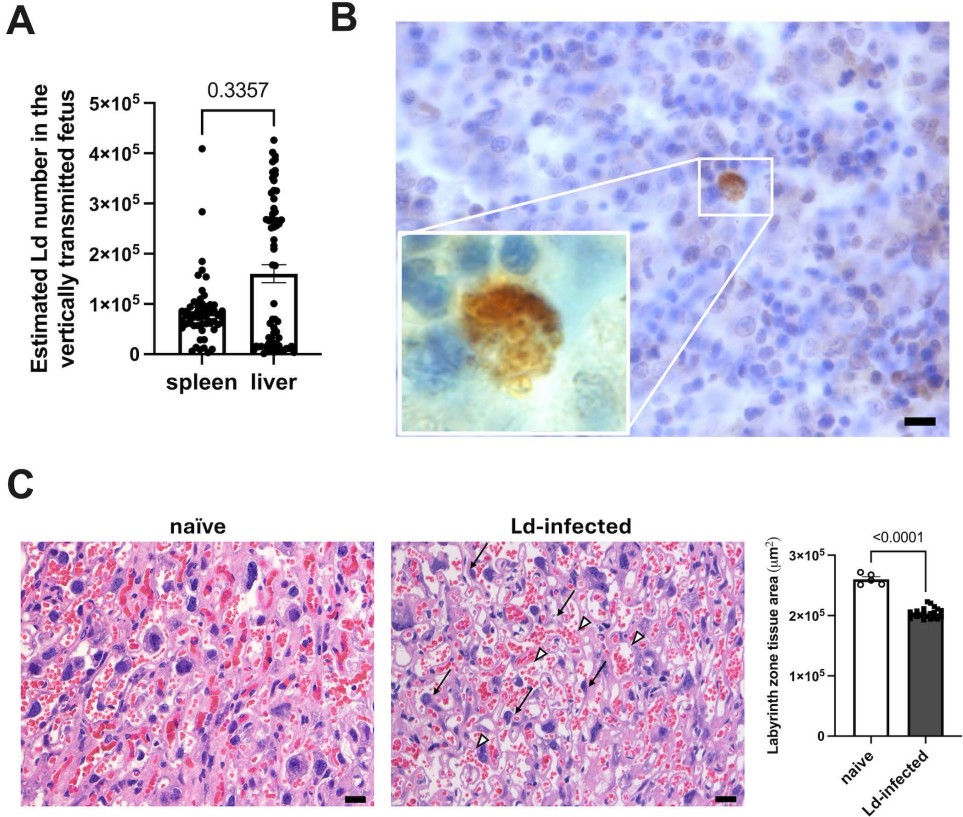

**Fig 4. Vertical transmission of Ld with placental degeneration (A)** Estimated Ld number in the spleens and livers of vertically transmitted fetuses (n = 58-65). Means ± SE are presented. *P* value for Mann-Whitney test is shown. **(B)** Detection of KMP11-positive cells in fetal liver. **(C)** Left: Ld-induced placental degeneration. Trophoblastic cell atrophy (arrow) and vasodilation (arrowhead) were observed in HE-stained infected placentas. Bars, 20 μm. Right: Labyrinth zone tissue area excluding vascular lumen at 200 × magnification (n = 5-21). Means ± SE are presented. *P* value for Student's t test is shown.

while there was no change in the number of cNK cells, the number of uNK cells tended to increase in the placental LZ of the Ld-infected group (Fig 5C). The results showed that Ld infection induced placental villitis with T cell infiltration.

Placental degeneration as described above was observed in all placentas of Ld-infected mice regardless of the presence or absence of vertical transmission, and no correlation was observed between the number of infiltrating cells and vertical transmission, although CD206⁺ MΦ and uNK cells tended to increase slightly in fetuses with vertical transmission (S1 Fig). The results suggest that Ld infection weakens the placental barrier by inducing placental degeneration, and the Ld parasites that are able to pass through the barrier may contribute to vertical transmission.

**Ld infection induces activation of IFN signals in the placenta.**

Finally, RNA-Seq analysis of the placenta was performed to extract molecules related to placental immunopathology in Ld-infected mice. The results identified 82 differentially expressed genes in the placentas of Ld-infected mice compared to naïve controls (S1 Table). Additionally, 61 genes thought to be involved in placental pathology were extracted based on gene function (Table 3). Among them, 29 genes involved in IFN signaling accounted for the most. The identified IFN signal-related genes include genes involved in the induction of IFN (increase in *Parp9* [39,40], *Nod2* [41,42] and decrease in *Ddah2* [43]), genes involved in IFN signal transduction (increase in *Stat1* [44], *Stat2* [44], *Irf1* [45,46]), and genes whose

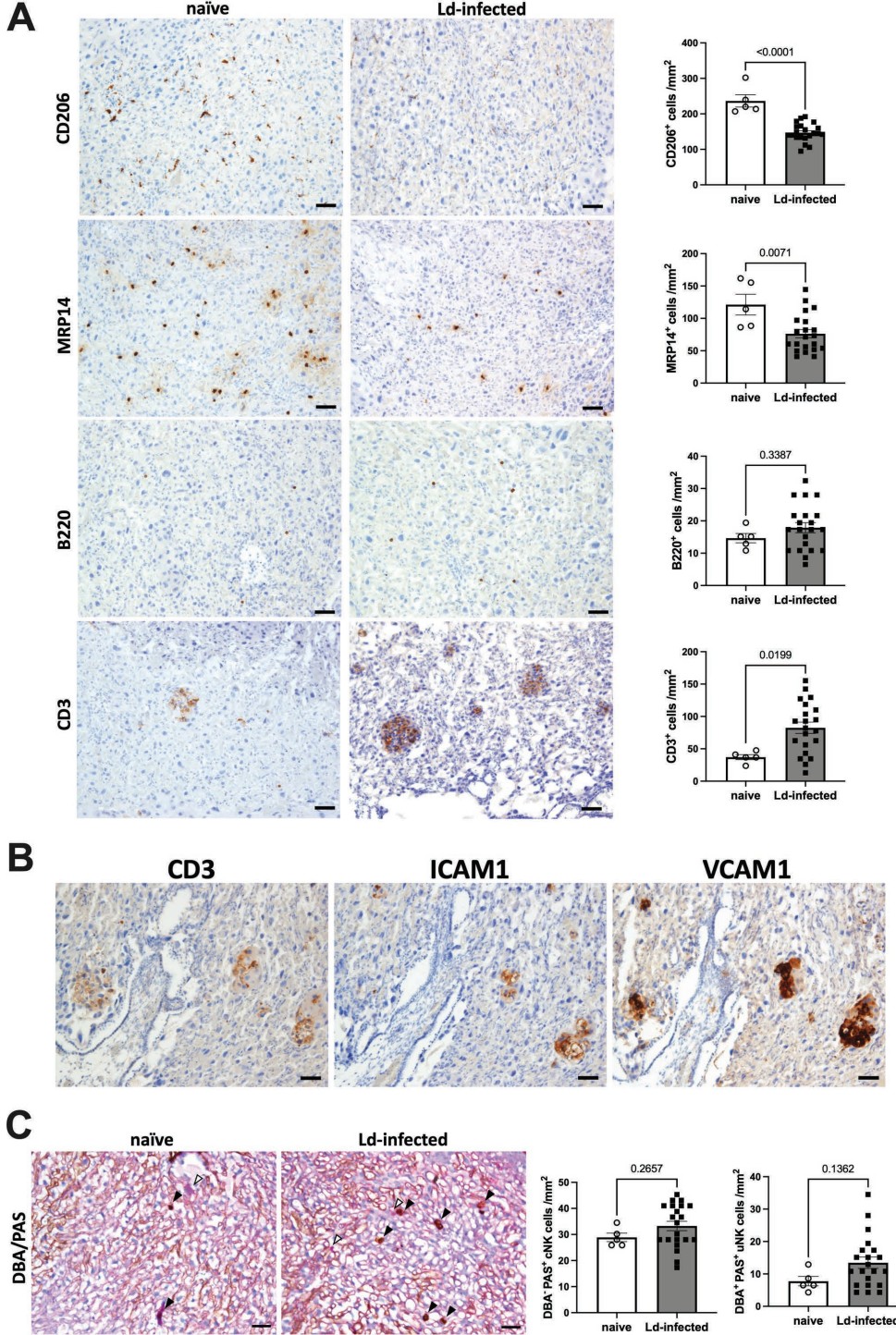

**Fig 5. Villitis with T cell infiltration in the placenta of Ld-infected mice. (A)** The expression of CD206, MRP14, B220 and CD3 in the placentas. Ld infection decreased CD206⁺ MΦs and MRP14⁺ MΦs, and increased CD3⁺ T cells in the placentas. **(B)** Consistent localization of CD3, ICAM1, and VCAM1 in serial placental sections. **(C)** cNK and uNK cells in DBA/PAS-stained placental sections. Open arrowhead indicates cNK cells (DBA⁻ PAS⁺), and closed arrowhead indicates uNK cells (DBA⁺ PAS⁺). The number of cells was counted in 5 random microscopic fields of placental labyrinth zone at 400× magnification (n = 5-21). Bars, 50 μm. Means ± SE are presented. *P* values for Student's t test are shown.

expression is induced by IFN (IFN-stimulated genes, ISGs; increase in *Apol11b* [47], *Iigp1c* [48], *Zbp1* [49], *Iigp1* [48], *Igtp* [50], *Ifi44l* [44,46], *Ifit1bl2* [44,46], *Gbp2b* [51], *Irgm1* [52], *Gbp2* [48], *Gbp3* [48], *Ifit1* [44,46], *Irgm2* [53], *Ifit2* [44,46], *Dtx3l* [54], *Rtp4* [55], *Trim21* [56], *Oasl2* [44,46], *Uba7* [57], *Parp10* [58], *Samhd1* [59], *Oas1b* [44,46]). Among the ISGs, *Iigp1c*, *Iigp1*, *Igtp*, *Gbp2b*, *Irgm1*, *Gbp2*, *Gbp3* and *Irgm2* were IFN-induced GTPases. In addition, a decrease in *Fcrl6*, whose expression induces IFN-γ promotion, was also observed [60]. The second most abundant gene group included 18 genes involved in MHC I antigen recognition induced downstream of IFN signals [61]. In brief, Ld infection induced the increase in *Cd3e* (which is involved in TCR-mediated signal transduction from MHC [62]), *Tap1* (which mediates the presentation of degraded peptide fragments from the endoplasmic reticulum to MHC I [63]), *Nlrc5* (a transactivator of MHC I [64]), *Psmb9* (which is involved in the antigen degradation pathway [65,66]), and MHC I components (*H2-Q5, H2-Q6, H2-Q7, H2-Q10, B2m, H2-Q2, H2-T22, H2-D1, H2-K2, H2-T26, H2-T7, H2-Q1, H2-Q4 and H2-K1*).

On the other hand, in the placentas of the Ld-infected group, the increase in genes involved in cellular immune suppression and inflammation control was also observed. Genes involved in cellular immunosuppression include the increase in *PD-L1* (which is involved in suppressing T cell effector responses [67]), the decrease in granzyme and perforin (*Gzme, Gzmc, Gzmd, Gzmb, Gzmg, Gzmf* and *Prf1*), and the decrease in *Zbtb32* (a negative regulator of the transcriptional regulator GATA3 [68], which acts on Th2 induction and Th1 suppression [76]). Genes related to inflammation control included the increase in *Psg16* and *Pla2g7* (which induce activation of anti-inflammatory MΦ [69,70]), and the increase in *Trip10* (which is important for VCAM1/ICAM1-mediated T cell adhesion and infiltration [71], and also has an inhibitory effect on IFN-γ signaling [72]). In addition, *Ctsg*, which induces the activation of T cells and NK cells [73,74], was decreased in the placentas of the Ld-infected group. Furthermore, the decrease in *Pdgfb*, a growth factor important for the development of the placental labyrinth layer [75], was also observed. The results indicate that Ld infection induces the activation of IFN signals simultaneously with cellular immune suppression and inflammation control in the placentas.

## Discussion

Here, we elucidated the effects of maternal chronic VL on maternal-fetal immunopathology. Our model showed that pregnancy did not worsen the maternal VL symptoms of anemia and splenomegaly (Fig 1A and 1B). Moreover, pregnancy did not affect the number of Ld parasites in the maternal spleens and livers (Fig 1C). Pregnancy also did not exacerbate the levels of BUN, ALT, and IFN-γ, which are increased by Ld infection (Fig 2A). The results show that pregnancy does not worsen maternal chronic VL symptoms.

Rather, in our model, pregnancy tended to alleviate maternal chronic VL symptoms. For example, the hematological parameters of Ld-infected dams showed a tendency to recover during pregnancy. The phenomenon may be related to the promotion of hematopoiesis induced by hormones and cytokines as a normal response to pregnancy [77–79]. In addition, serum BUN levels in the pregnant infected group were reduced to the same level as in pregnant naïve controls, which is consistent with the general fact that BUN decreases during pregnancy due to increased renal blood flow and malnutrition [80]. Furthermore, pregnancy also suppressed the increase in serum IFN-γ induced by Ld infection (Fig 2B). One possible reason for this is that pregnancy hormones induce Th1 suppression during pregnancy to maintain immune tolerance to the fetus [81]. Liver weight increased with pregnancy in both naïve and infected groups (Fig 1B), which is also a normal response to pregnancy [82].

Our results pose important issues in VL during pregnancy. The first issue is that the symptoms of pregnancy and VL are so similar that it is difficult to differentiate them based on the symptoms alone. As shown above, VL symptoms such as anemia and hepatomegaly are also observed in normal pregnancies (Fig 1A and 1B). To definitively diagnose VL from pregnancy, it is necessary to detect *Leishmania* parasites or specific antibodies against them. However, if pregnant patients misunderstand VL symptoms as pregnancy symptoms, they will not visit a medical institution for treatment. The second issue is that even if pregnant VL patients are able to access medical institutions, the pregnancy reaction masks the actual VL symptoms, making it difficult for medical doctors to accurately assess the VL pathology and overlooking the

**Table 3. RNA-Seq transcriptome analysis of immune factors in placentas of pups born to Ld-infected dams.**

| gene name | log2 fold change (Ld/Naive) | adjusted *P* value | function | ref |
|---|---|---|---|---|
| *Parp9* | 0.82 | 1.E-07 | Induction of IFN | [39,40] |
| *Nod2* | 0.72 | 1.E-02 | Induction of IFN | [41,42] |
| *Ddah2* | -0.64 | 1.E-02 | Negative regulation of type I IFN induction | [43] |
| *Stat1* | 0.83 | 2.E-07 | IFN signaling | [44] |
| *Stat2* | 0.66 | 3.E-08 | IFN signaling | [44] |
| *Irf1* | 0.63 | 1.E-02 | IFN signaling | [45,46] |
| *Apol11b* | 5.23 | 3.E-11 | IFN-stimulated genes | [47] |
| *Iigp1c* | 2.47 | 1.E-13 | IFN-stimulated genes (GTPase) | [48] |
| *Zbp1* | 2.23 | 8.E-40 | IFN-stimulated genes | [49] |
| *Iigp1* | 2.19 | 2.E-05 | IFN-stimulated genes (GTPase) | [48] |
| *Igtp* | 1.80 | 3.E-08 | IFN-stimulated genes (GTPase) | [50] |
| *Ifi44l* | 1.46 | 1.E-04 | IFN-stimulated genes | [44,46] |
| *Ifit1bl2* | 1.34 | 4.E-02 | IFN-stimulated genes | [44,46] |
| *Gbp2b* | 1.11 | 1.E-04 | IFN-stimulated genes (GTPase) | [51] |
| *Irgm1* | 1.10 | 7.E-06 | IFN-stimulated genes (GTPase) | [52] |
| *Gbp2* | 1.05 | 1.E-04 | IFN-stimulated genes (GTPase) | [48] |
| *Gbp3* | 0.99 | 2.E-03 | IFN-stimulated genes (GTPase) | [48] |
| *Ifit1* | 0.98 | 1.E-03 | IFN-stimulated genes | [44,46] |
| *Irgm2* | 0.94 | 1.E-04 | IFN-stimulated genes (GTPase) | [53] |
| *Ifit2* | 0.90 | 3.E-02 | IFN-stimulated genes | [44,46] |
| *Dtx3l* | 0.86 | 8.E-05 | IFN-stimulated genes | [54] |
| *Rtp4* | 0.77 | 8.E-05 | IFN-stimulated genes | [55] |
| *Trim21* | 0.76 | 2.E-06 | IFN-stimulated genes | [56] |
| *Oasl2* | 0.72 | 1.E-03 | IFN-stimulated genes | [44,46] |
| *Uba7* | 0.65 | 8.E-05 | IFN-stimulated genes | [57] |
| *Parp10* | 0.61 | 3.E-11 | IFN-stimulated genes | [58] |
| *Samhd1* | 0.58 | 4.E-04 | IFN-stimulated genes | [59] |
| *Oas1b* | 0.58 | 9.E-03 | IFN-stimulated genes | [44,46] |
| *Fcrl6* | -0.83 | 5.E-02 | Negative regulation of IFN-γ induction | [60] |
| *Cd3e* | 1.90 | 3.E-02 | MHC signaling | [62] |
| *Tap1* | 1.38 | 2.E-17 | MHC I antigen presentation | [63] |
| *Nlrc5* | 0.88 | 8.E-05 | Transactivator of MHC I | [64] |
| *Psmb9* | 0.83 | 1.E-02 | MHC I pathway antigen degradation | [65,66] |
| *H2-Q5* | 1.14 | 1.E-04 | MHC I components | |
| *H2-Q6* | 1.06 | 2.E-04 | MHC I components | |
| *H2-Q7* | 0.99 | 2.E-04 | MHC I components | |
| *H2-Q10* | 0.76 | 1.E-02 | MHC I components | |
| *B2m* | 0.75 | 5.E-03 | MHC I components | |
| *H2-Q2* | 0.69 | 1.E-02 | MHC I components | |
| *H2-T22* | 0.69 | 2.E-03 | MHC I components | |
| *H2-D1* | 0.67 | 5.E-03 | MHC I components | |
| *H2-K2* | 0.67 | 2.E-02 | MHC I components | |
| *H2-T26* | 0.63 | 5.E-03 | MHC I components | |
| *H2-T7* | 0.62 | 1.E-02 | MHC I components | |
| *H2-Q1* | 0.60 | 9.E-03 | MHC I components | |

*(Continued)*

**Table 3.** (Continued)

| gene name | log2 fold change (Ld/Naive) | adjusted *P* value | function | ref |
|---|---|---|---|---|
| *H2-Q4* | 0.60 | 3.E-03 | MHC I components | |
| *H2-K1* | 0.58 | 1.E-02 | MHC I components | |
| *CD274 (PD-L1)* | 1.05 | 6.E-06 | Suppression of T cell effector responses | [67] |
| *Gzme* | -0.85 | 6.E-03 | Granzyme | |
| *Prf1* | -0.82 | 6.E-03 | Perforin | |
| *Gzmc* | -0.79 | 1.E-02 | Granzyme | |
| *Gzmd* | -0.73 | 2.E-02 | Granzyme | |
| *Gzmb* | -0.71 | 3.E-02 | Granzyme | |
| *Gzmg* | -0.68 | 4.E-02 | Granzyme | |
| *Gzmf* | -0.65 | 2.E-02 | Granzyme | |
| *Zbtb32* | -0.69 | 5.E-02 | Induction of Th1 response | [68] |
| *Psg16* | 1.12 | 1.E-02 | anti-inflammatory MΦ activation | [69] |
| *Pla2g7* | 1.08 | 3.E-02 | anti-inflammatory MΦ activation | [70] |
| *Trip10* | 0.67 | 8.E-03 | T cell infiltration and IFN-γ signaling suppression | [71,72] |
| *Ctsg* | -0.86 | 2.E-02 | Activation of NK cell and T cell | [73,74] |
| *Pdgfb* | -0.76 | 4.E-02 | Placental labyrinth layer development | [75] |

patients. As we have shown above, serum BUN, ALT, and IFN-γ, which are normally elevated in VL, are suppressed in pregnant VL. Serum IFN-γ has been proposed as an indicator of active VL [83], but its application may be difficult in pregnant women. Therefore, the results suggest the necessity for diagnostic criteria specific to pregnant VL.

In contrast, some cases reported that pregnancy worsens the symptoms of leishmaniasis [16,24,25,84,85]. Relatedly, pregnancy impairs resistance and worsens skin lesions of C57BL/6 mice, which are normally resistant to *L. major* infection [25,85]. The opposite effect of pregnancy on leishmaniasis symptoms may be related to the following factors including parasite species, host resistance/susceptibility, timing of infection, and pregnancy period. In *L. major* infection, the Th1/Th2 paradigm is well known, in which Th1-type immunity contributes to resistance and Th2-type immunity contributes to susceptibility [86,87]. Here, we used susceptible BALB/c mice to elucidate the effects of maternal chronic VL on maternal-fetal immunopathology. On the other hand, considering that Th1 suppression is induced during pregnancy [81], it can be assumed that the worsening of VL symptoms due to pregnancy is more pronounced in resistant hosts. Therefore, it is conceivable that some patients may be susceptible to VL during pregnancy, even though they are normally resistant. In addition, the timing of infection is an important factor. In this study, we used dams in the chronic phase of infection, which may have made it difficult to observe differences in symptoms. In contrast, during the acute phase of infection, pregnancy might have a significant impact on VL symptoms. Therefore, the effects of pregnancy on VL symptoms may differ between the acute and chronic stages of infection. Furthermore, the pregnancy period is also an important point. Because the human pregnancy period (266 days) is longer than that of mice (20 days), human mothers and fetuses should suffer long-term effects of pregnant VL. Therefore, to better understand pregnancy VL in humans, other animal models with longer gestation periods, such as dogs and monkeys, may also need to be considered.

Regarding the effects on pregnancy, Ld infection was shown to reduce mating success rate and pregnancy rate (Fig 3A and Table 1). Because it has been reported that *Leishmania* infection tends to reduce sex hormone levels [88], disturbances in sex hormones may be involved in the decreased mating success rate due to Ld infection. Additionally, consistent with our results, it has been reported that VL pregnant women have an increased risk of miscarriage [16,18] and that *L. major* and *L. amazonesis* infections reduce conception rates in mice [89,90]. This may be due to the increase in IFN-γ

caused by Ld infection. For example, it has been reported that decreased conception rates due to *L. major* infection is correlated with increased IFN-γ [89]. Increased IFN-γ has also been shown to be associated with decreased conception rates in other intracellular parasitic infections such as malaria [91,92] and toxoplasmosis [93]. In fact, our results showed that serum IFN-γ in pregnant Ld-infected mice was at least 5 times higher than in the pregnant naïve group, although it was suppressed compared to non-pregnant infected controls (Fig 2B). Therefore, IFN-γ is considered to be one of the factors contributing to the decreased pregnancy rate in VL.

Furthermore, fetal developmental delay and tissue degeneration have also been reported in pregnant leishmaniasis [18,19,28,94]. On the other hand, in our model, when Ld infection did not cause miscarriage, there was no apparent effect on fetal number or development (Fig 3C). In a hamster model, it has been reported that fetal mortality and weight gain rates due to Ld infection vary depending on whether the pregnancy is during the acute or chronic infection stage [94], suggesting that the effect on the fetus depends on when the mother was infected with Ld. Similar to our results, several human cases have been reported in which VL mothers gave birth to newborns with no abnormalities immediately after birth [18]. However, even in such cases of normal birth, vertical transmission to the child may only become apparent several months after birth [18]. Therefore, we investigated whether maternal Ld infection causes vertical transmission to the fetuses.

As a result, *Leishmania* DNA was detected in the fetuses (Fig 4A and Table 2), and Ld-infected cells were confirmed in the fetal livers (Fig 4B). The results demonstrated vertical transmission in Ld-infected mice. We confirmed that DNA samples extracted from 100 μl blood of Ld-infected dams were PCR negative for LSU rRNA gene, indicating that the Ld concentration in maternal blood is at least lower than 1 parasite/1 μl. Therefore, it can be said that maternal blood contamination has little effect on the PCR results for the fetal LSU rRNA gene. Previous studies have also demonstrated vertical transmission in rodent VL models such as *L. infantum*-infected BALB/c mice [95] and *L. donovani*-infected hamsters [94]. Additionally, *L. panamensis* and *L. mexicana*, which cause cutaneous leishmaniasis, have also been reported to cause vertical transmission in hamsters and mice [28,94]. As shown in Table 2, Our model showed higher vertical transmission rates compared to previous VL models (4.5–25.8%) [94,95]. However, despite histological confirmation of *Leishmania* parasites in the placenta in human and dog case reports [29,96], very few Ld parasites were found in the placentas of Ld-infected mice. Because humans and dogs have longer pregnancy periods than mice, it is thought that *Leishmania* parasites that reach the placenta have enough time to multiply. There are limited reports of histological analysis of the placentas of pregnant VL rodents, and further research is required.

Importantly, vertical transmission occurs even though few typical Ld-infected cells are observed histologically in the placenta. The reason is thought to be the presence of trace amounts of Ld in the blood [2]. The results suggest that even if *Leishmania* parasites are not detected in the placentas, the congenital infection risk of babies born to VL mothers is not excluded. In fact, vertical transmission from asymptomatic VL mothers has also been reported [97]. Our results also indicate that even when Ld is vertically transmitted, no obvious congenital abnormalities may be observed (Fig 3C). Therefore, long-term follow-up is necessary to check whether offspring born to VL mothers develop the VL after birth.

Pathological analysis of the placenta revealed placental degeneration of trophoblast cell atrophy and vasodilation due to Ld infection (Fig 4C). Possible factors of vascular dilation in the placenta include nitric oxide (NO), prostaglandin E2 (PGE2), vascular endothelial growth factor (VEGF), and hypoxic conditions. RNA-seq data from Ld-infected placentas showed no significant changes in the *Nos2*, *Ptges2*, and VEGF genes (S2 Table). On the other hand, in Ld-infected placentas, genes related to oxidative phosphorylation, including *Atp5g3*, *Atp5g1*, *Ndufb7*, *Ndufa4*, *Atp5e*, and *Uqcrc1*, were downregulated, while the *Arntl2* gene, which is induced in response to hypoxia [98], was upregulated (S1 Table). Consistent with this, it has been reported that hypoxic conditions are induced during Ld infection [99]. These results suggest that the hypoxic environment induced by Ld infection in the placenta may contribute to vascular dilation.

In addition, villitis accompanied by T cell infiltration was observed, and the affected areas corresponded with the localization of ICAM1 and VCAM1 (Fig 5). Since IFN-γ has been reported to induce the expression of ICAM1 and VCAM1

[100,101], it is possible that the increased levels of IFN-γ during Ld infection contribute to the upregulation of ICAM1 and VCAM1 expression in the placenta. Similarly, trophoblast cell necrosis of placental villi has been reported in severe cases of pregnant VL in humans and dogs [29,96]. In this context, placental pathology in other protozoal infections which induce vertical transmission (such as malaria, toxoplasmosis, and trypanosomiasis) is characterized by villous degeneration with proliferation of parasites and MΦs in the placental tissues [102]. Conversely, in the placentas of Ld-infected mice, MΦs decreased while T cells increased (Fig 5). Importantly, our results showed that placental tissue degeneration occurs even though few *Leishmania* parasites are identified in the placentas (Figs 4 and 5). Relatedly, it has been reported that non-infectious factors such as obesity induce placental tissue degeneration such as trophoblast cell necrosis via elevated maternal inflammatory cytokines [103,104], and maternal anemia also induces reorganization of placental structure and angiogenesis [105,106]. Additionally, disruption of the placental barrier due to placental degeneration is associated with vertical transmission [102,107]. Therefore, it is suggested that elevated inflammatory mediators and anemia caused by maternal Ld infection induce the placental tissue degeneration, which contributes to vertical transmission. The present study elucidated the unique placental histopathology of VL, which is different from other protozoal infections. In the future, it is expected that our pregnant VL model will be utilized to further understand the mechanism of maternal-fetal immuno-pathology caused by VL.

Furthermore, RNA-Seq analysis of the placenta revealed that Ld infection promotes genes involved in the activation of IFN signal in the placentas. On the other hand, Ld infection also induced genes involved in cellular immunosuppressive signals such as granzyme reduction and T cell suppression in the placentas (Table 3). While IFN-γ is upregulated during Ld infection (Fig 2B), accumulating evidence indicates that type I interferons also contribute to the immunopathology of VL. For example, dsRNA viruses that infect *Leishmania* parasites [108] and NOD2 receptors [41] increased in the VL placenta have been shown to induce type I IFN. In addition, TNF-α and IFN-γ, which are increased by Ld infection, also induces NOD2 [109], suggesting that the inflammatory cytokines may also be involved in inducing IFN signaling through promotion of NOD2 expression in VL placentas.

Although we did not measure serum levels of type I IFN in this study, bulk RNA-seq analysis of the placenta, spleen, and liver from Ld-infected mice showed no upregulation of type I IFN expression (S3 Table). Similarly, type I IFNs have not been detected in the serum of human VL patients [110]. However, RNA-seq analysis of peripheral blood leukocytes from VL patients has shown activation of type I IFN signaling, suggesting a potential involvement in disease progression [111]. These findings indicate that even when type I IFN levels are low in serum or whole tissues, IFN signaling can be activated locally in specific sites, and the placenta is thought to be one such example. Indeed, the placenta is a unique immune environment in which IFN signaling is more likely to be activated [112]. Furthermore, it is known that interactions with toll-like receptors and the STING pathway can enhance IFN signaling [113,114], which may also contribute to the localized activation. In addition, an IFN-independent ISG induction pathway has also been reported in leishmaniasis [108]. Taken together, these findings suggest that even in the absence of detectable type I IFN in serum, local activation of IFN signaling in the placenta is entirely plausible. This may be attributed to the placenta's unique immune environment and the involvement of diverse signaling pathways.

It is important to note that many of the genes upregulated in the Ld-infected placenta may be induced not only by type I IFN but also by IFN-γ. Furthermore, considering that IFN-γ levels increase in the serum during Ld infection (Fig 2B), it is difficult to conclude that these changes are solely due to type I IFN. The potential involvement of IFN-γ in the disease pathology must also be taken into account. Although IFN-γ is a central mediator of the Th1 immune response, its excessive activation has been associated with immune-mediated organ damage and chronic inflammation in VL [134]. Therefore, in pregnant VL, both type I IFN and IFN-γ may be intricately involved in the pathogenesis of placental inflammation, and further experimental studies are needed to clarify their respective roles.

In chronic inflammation, IFN promotes PD-L1 expression, thereby inducing polarization of anti-inflammatory MΦ/DC [115–118], and inhibiting cytotoxic T cells [119]. Additionally, MHC I, whose expression is promoted downstream of IFN

signals [61], suppresses the cytotoxic activity of NK cells [120,121]. In this way, IFN have been shown to induce functional suppression of T cells and NK cells, leading to failure of infection control and worsening of disease [122,123]. Similarly, in Ld-infected mice, type I IFN has been shown to induce Th1 immunosuppression and Th2 immune activation, contributing to VL susceptibility [108,111]. Also in human VL patients, it has been reported that type I IFN signals are activated and are involved in disease aggravation [111]. Consistently, in Ld-infected placentas, changes in the expression of genes involved in T cell suppression (*PD-L1, Zbtb32, Trip10* and *Ctsg* [68,71–74,124]), anti-inflammatory MΦ polarization (*Psg16* and *Pla2g7* [69,70]), and NK cell suppression (*Ctsg* [73,74], granzymes and perforin) were observed at the same time as IFN signal activation. Among the genes, PD-L1 has been reported to contribute to *Leishmania* susceptibility [125–127]. Therefore, it is thought that the infiltrated T cells observed in the Ld-infected placentas (Fig 5) are also suppressed by IFN. Moreover, it should be taken into consideration that cellular immunity is further suppressed by hormonal effects during pregnancy [128]. Suppression of cellular immunity in the placentas may provide a favorable environment for *Leishmania* parasite survival and facilitate vertical transmission. Besides, because IFN also induces MΦ apoptosis in *M. tuberculosis* infection [129,130], IFN may be one of the factors responsible for the decrease in MΦs in Ld-infected placentas (Fig 5A). In addition, high levels of IFN have been reported to contribute to placentitis and impaired placental development [131,132]. Relatedly, the NOD2 receptor, which is involved in type I IFN induction, contributes to abortion due to *Brucella abortus* infection [133]. The results suggest that activation of IFN signals in the VL placentas induces cellular immunosuppression and placental degeneration, contributing to vertical transmission.

In conclusion, our study revealed that chronic Ld infection induces decreased pregnancy rate, vertical transmission, placental degeneration with T cell infiltration, and IFN signal activation in the placentas. The results suggest that IFN induces cellular immunosuppression (providing an environment favorable for protozoan survival) and placental degeneration, which contribute to vertical transmission.

Importantly, maternal inflammatory responses during pregnancy, with or without vertical transmission, have lifelong effects on offspring immune development [135–138]. In fact, it has been reported that offspring born to *Leishmania*-infected mothers exhibit increased susceptibility to *Leishmania* infection [21,94,139]. We also observed a reduction in the number of fetal MΦs associated with changes in maternal cell populations in Ld-infected placentas, independent of the presence or absence of vertical transmission (Figs 5A and S1). The results suggest that the maternal immune response to infection influences the immune development of the offspring also in VL. Therefore, it is necessary to evaluate the effects of pregnant VL on offspring not only during the perinatal period but also over the long term after birth.

Pregnant VL can have diverse effects on mother and child depending on complex factors such as parasite species, host resistance, timing of infection, and pregnancy period. However, there are currently limited reports to understand its pathogenesis. In order to gain a deeper understanding of the actual situation of pregnant VL, further research including large-scale epidemiological surveys is required in parallel with pathological analysis using animal models.

## Supporting information

**S1 Fig. Little relationship between placental cell infiltration and vertical transmission.** The number of infiltrating cells in the placentas of *Leishmania* gene-negative and positive fetuses. The number of cells was counted in 5 random microscopic fields of placental labyrinth zone at 400×magnification (n=3–7). Means±SE are presented. *P* values for Student's t test are shown.
(PDF)

**S1 Table. Differentially expressed genes in Ld-infected placenta.**
(PDF)

**S2 Table. Genes related to vasodilation in Ld-infected placenta.**
(PDF)

**S3 Table. IFN genes in Ld-infected placenta, spleen and liver.**
(PDF)

## Author contributions

**Conceptualization:** Haruka Mizobuchi.

**Data curation:** Haruka Mizobuchi, Junya Yamagishi.

**Formal analysis:** Haruka Mizobuchi, Junya Yamagishi.

**Funding acquisition:** Haruka Mizobuchi, Yasuyuki Goto.

**Investigation:** Haruka Mizobuchi.

**Methodology:** Haruka Mizobuchi.

**Project administration:** Haruka Mizobuchi, Yasuyuki Goto.

**Resources:** Haruka Mizobuchi, Yasuyuki Goto.

**Supervision:** Chizu Sanjoba, Yasuyuki Goto.

**Validation:** Haruka Mizobuchi.

**Visualization:** Haruka Mizobuchi.

**Writing – original draft:** Haruka Mizobuchi.

**Writing – review & editing:** Chizu Sanjoba, Yasuyuki Goto.

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
