## [Decision Letter · Decision Letter 0]

Response to Reviewers
Revised Manuscript with Track Changes
Manuscript

We look forward to receiving your revised manuscript.

Shaden Kamhawi

co-Editor-in-Chief

Paul Brindley

co-Editor-in-Chief

**Additional Editor Comments :**
**Journal Requirements:**

At this stage, the following Authors/Authors require contributions: Haruka Mizobuchi, Junya Yamagishi, Chizu Sanjoba, and Yasuyuki Goto. Please ensure that the full contributions of each author are acknowledged in the "Add/Edit/Remove Authors" section of our submission form.

- ® on page: 10

- TM on pages: 8, and 9.

3) Tables should not be uploaded as individual files. Please remove these files and include the Tables in your manuscript file as editable, cell-based objects. For more information about how to format tables, see our guidelines:

https://journals.plos.org/plosntds/s/tables 

4) Please amend your detailed Financial Disclosure statement. This is published with the article. It must therefore be completed in full sentences and contain the exact wording you wish to be published.

2) State what role the funders took in the study. If the funders had no role in your study, please state: "The funders had no role in study design, data collection and analysis, decision to publish, or preparation of the manuscript.".

**Reviewers' comments:**

**Key Review Criteria Required for Acceptance?**

**Methods**

-Are the objectives of the study clearly articulated with a clear testable hypothesis stated?

-Is the study design appropriate to address the stated objectives?

-Is the population clearly described and appropriate for the hypothesis being tested?

-Is the sample size sufficient to ensure adequate power to address the hypothesis being tested?

-Were correct statistical analysis used to support conclusions?

-Are there concerns about ethical or regulatory requirements being met?

Reviewer #1: Fig. 4B. It is not clear how may mice/organs/sections were evaluated to conclude that KMP11+ cells are detectable in fetal liver, but ‘only a few KMP11+ cells were observed in the placentas’, especially when Fig. 4B only shows a single positive cell in the image. Quantification of microscopy would strengthen the observation of this novel finding.

Line 307: Quantification is needed to support ‘the density of the LZ was lower than that of naïve controls’.

Was IFNa/b or IFNg increased in RNASeq data? IFNg is decreased with pregnancy in infected mice, but it is still elevated compared to naïve/pregnancy or infected/non pregnant. Similarly, were type I IFNs elevated in the serum like IFNg in Fig. 2B?

Fig. 5C, lines 324-325. The data does not support more uNK cells, so the emphasis should be decreased on their infiltration in line 325, 467, and 473.

Describe how estimate Ld number in fetus was calculated in M&M (Fig. 4A).

**Results**

-Does the analysis presented match the analysis plan?

-Are the results clearly and completely presented?

-Are the figures (Tables, Images) of sufficient quality for clarity?

Reviewer #1: Discussion: What causes the vessel diameter? Are vascular mediators like VEGF-A elevated in the RNASeq dataset that couple help explain the dilation?

Discussion: Is elevated ICAM and VCAM due to elevated IFNg?

Discussion: What restricts the CD3 T cells from moving throughout the tissue or why or the cells cluster?

Line 335-336: no control group is mentioned. Placentas of Ld infected compared to...

Line 448: Is BALA/c a typo?

Line 451 need a citation

**Conclusions**

-Are the conclusions supported by the data presented?

-Are the limitations of analysis clearly described?

-Do the authors discuss how these data can be helpful to advance our understanding of the topic under study?

-Is public health relevance addressed?

Reviewer #1: Many genes mentioned in Table 3 are induced by either type I IFNs or IFNg, using STAT1 as a prime example. Therefore, the conclusion that type I IFNs are solely responsible is not obvious or supported without pathway analysis or uncoupling the roles of each pathway through blockade strategies.

To make the conclusion that barrier integrity is affected, the authors would need to assess barrier integrity using a common assay like evans blue which measures leakiness, or a comparable approach.

**Editorial and Data Presentation Modifications?**

Reviewer #1: Arrows denoting LZ, trophoblast atrophy, and vasodilation would help the reader for Fig. 4C. For instance, it should be made clear if the nuclei are of villi or trophoblasts?

**Summary and General Comments**

Reviewer #1: This study shows that chronic Ld infection induces decreased pregnancy rate, vertical transmission, placental degeneration with T cell infiltration. The manuscript is well-written and addresses an important unanswered question in NTDs. Plus, the new experimental model developed will be informative for future studies dissecting the immunopathology associated with pregnancy in VL. However, the conclusion that type I IFN signaling in the placentas is responsible for the phenotype is less convincing as many of the targets identified by RNASeq can be induced by type I IFNs and IFNg. Therefore, the conclusion that type IFNs are important should be dampened (in the results, discussion, etc) as the current study does not uncouple IFNa/b from IFNg.

PLOS authors have the option to publish the peer review history of their article (what does this mean? ). If published, this will include your full peer review and any attached files.

**Do you want your identity to be public for this peer review?** For information about this choice, including consent withdrawal, please see our Privacy Policy .

Reviewer #1: No

**Figure resubmission:****Reproducibility:** To enhance the reproducibility of your results, we recommend that authors of applicable studies deposit laboratory protocols in protocols.io, where a protocol can be assigned its own identifier (DOI) such that it can be cited independently in the future. Additionally, PLOS ONE offers an option to publish peer-reviewed clinical study protocols. Read more information on sharing protocols at https://plos.org/protocols?utm_medium=editorial-email&utm_source=authorletters&utm_campaign=protocols

---

## [Editor Report · Decision Letter 1]

Dear Ms Mizobuchi,

We are pleased to inform you that your manuscript 'Vertical transmission of Leishmania donovani with placental degeneration in the pregnant mouse model of visceral leishmaniasis' has been provisionally accepted for publication in PLOS Neglected Tropical Diseases.

Best regards,

Peter E Kima

Guest Editor

Laura-Isobel McCall

Section Editor

Shaden Kamhawi

co-Editor-in-Chief

Paul Brindley

co-Editor-in-Chief

---

## [Editor Report · Acceptance letter]

Dear Ms Mizobuchi,

We are delighted to inform you that your manuscript, "Vertical transmission of Leishmania donovani with placental degeneration in the pregnant mouse model of visceral leishmaniasis," has been formally accepted for publication in PLOS Neglected Tropical Diseases.

Best regards,

Shaden Kamhawi

co-Editor-in-Chief

Paul Brindley

co-Editor-in-Chief
